# Genetic Profile and Toxigenic Potential of *Bacillus cereus* Isolates from a Norwegian Ice Cream Production Plant

**DOI:** 10.3390/foods13193029

**Published:** 2024-09-24

**Authors:** Toril Lindbäck, Ann-Katrin Llarena, Stine Göransson Aanrud, Marte Monshaugen, Yohannes B. Mekonnen, Carina Wiker Holmemo, Marina Aspholm

**Affiliations:** 1Unit of Food Safety, Department of Paraclinical Sciences, Norwegian University of Life Sciences, 1433 Ås, Norwaymarina.aspholm@nmbu.no (M.A.); 2Toxicology Unit, Department of Paraclinical Sciences, Norwegian University of Life Sciences, 1433 Ås, Norway; stine.goransson.aanrud@nmbu.no; 3MiraiFood Holmemo, Neuberggata 6D, 0367 Oslo, Norway; carina_wiker@hotmail.com

**Keywords:** *Bacillus cereus*, cereulide, emetic toxin, persistence, *Bacillus weihenstephanensis*, ice cream

## Abstract

Members of the *B. cereus* group are spore-forming organisms commonly associated with spoilage of milk and dairy products. We have determined the genetic identity and growth characteristics of 57 *B. cereus* isolates collected from a Norwegian ice cream production plant. Our findings revealed persistence of *B. cereus* spp. strains for up to 19 months, suggesting the plant’s susceptibility to long-term colonization. One of the mesophilic isolates, NVH-YM303, carried a complete cereulide synthetase operon. To assess the potential food poisoning risk associated with the presence of cereulide-producing strains in the production line, we examined the production of cereulide in ice cream and milk at different temperatures by NVH-YM303 and by the emetic psychrotrophic *B. weihenstephanensis* strain BtB2-4. Our findings revealed that NVH-YM303 produced higher levels of cereulide in ice cream as compared to milk. Furthermore, it was observed that NVH-YM303 produced more cereulide in ice cream at 25 °C compared to 15 °C. Conversely, BtB2-4 produced more cereulide in ice cream at 15 °C than at 25 °C. The results obtained in this study contribute to knowledge important for risk assessment of the potential hazards posed by the presence of *B. cereus* within ice cream production facilities.

## 1. Introduction

*Bacillus* spp. constitute a substantial part of the microbial community found in raw bovine milk. These bacteria can survive the pasteurization process in the form of spores, which poses a contamination problem for the dairy industry [1,2,3]. The viable spores can undergo germination and produce vegetative cells within dairy products. Assuming the temperature is suitable, the bacteria can multiply and subsequently secrete enzymes and toxins, which presents a concern for both the quality and safety of dairy products. *B. cereus* can cause both foodborne infection and intoxication characterized by diarrheal and emetic symptoms, respectively. The symptoms of the diarrheal disease are caused by one or more of the following enterotoxins: hemolysin BL (Hbl) [4], non-hemolytic enterotoxin (Nhe) [5], or Cytotoxin K (CytK) [6]. The emetic disease is, on the other hand, caused by the heat- and acid-stable cyclic dodecadepsipeptide [(D-O-Leu-D-Ala-L-O-Val-L-Val)_3_] toxin cereulide which is pre-formed in the food [7]. Symptoms caused by cereulide intoxication range from nausea and emesis to multi-organ failure, and even fatalities have been reported [8,9,10,11,12]. Cereulide is synthesized non-ribosomally by the multienzyme cereulide synthetase complex CesNRPS [13], and the genes encoding CesNRPS are carried by mega-plasmids resembling the *Bacillus anthracis* toxin plasmid pX01 [13,14].

*B. cereus* spp. strains which grow at 7 °C but not at 43 °C are considered psychrotrophic and species exhibiting these growth characteristics combined with the presence of the gene encoding cold-shock protein CspA have been defined as *Bacillus weihenstephanensis* [15,16]. However, bacterial taxonomy is changing due to whole genome sequencing, and a reclassification of *B. weihenstephanensis* to *Bacillus mycoides* has been suggested [17]. *B. weihenstephanensis* carrying CesNRPS encoding genes have also been described [18,19,20], and such strains pose a special hazard if they are able to produce cereulide at refrigerator temperatures. Jovanovic et al. (2022) reported that 12% of the psychrotrophic strains isolated from different types of foods carried the CesNRPS encoding genes [21]. While certain strains were able to produce cereulide at a temperature of 37 °C, none of them produced the toxin at 4 °C [21]. Similarly, Guérin et al. (2017) demonstrated that two emetic *B. weihenstephanensis* strains produced a low level of cereulide at 8 °C, indicating that the risk of emetic poisoning from food stored below 8 °C is low [22]. Additionally, the absence of a connection between *B. cereus* food poisoning and appropriately stored commercially produced minimally processed chilled foods may indicate that psychrotrophic strains exhibit limited pathogenicity [23].

Ice cream mass provides an ideal environment for growth of *B. cereus* due to its high nutrient content, neutral pH, and removal of the competing bacterial microbiota through pasteurization. And although ice cream has been the suspected food vehicle in outbreaks of foodborne disease, there are limited data available on the presence and potential growth of emetic *B. cereus* in ice cream mixture. It is reasonable to assume that if emetic *B. cereus* strains are left in tubes or equipment, especially during steps of the production process that occur at temperatures over 8 °C, the ice cream mixture may be at risk of containing cereulide.

The prevalence of intoxications caused by cereulide is expected to be quite common in Norway, and in other countries as well, but the exact incidence remains unknown. This is mostly because most cases are self-limiting, characterized by mild and temporary symptoms, and affected people rarely seek medical treatment. Moreover, food samples from outbreak situations are seldom investigated for the presence of emetic *B. cereus* or cereulide.

The objective of this study was to assess the food safety implications of *B. cereus* spp. growth in ice cream and dairy products. To achieve this, we collected 57 *B. cereus* isolates from the processing environment and products of a Norwegian ice cream producer. Through genome sequence analysis, we aimed to evaluate the virulence potential of these isolates and identify any persistent strains within the production facility. Additionally, we explored the temperature preferences of these isolates to determine optimal growth conditions. Furthermore, we tested the ability of two specific *B. cereus* spp. strains—*B. weihenstephanensis* BtB2-4, a psychrotrophic strain, and *B. cereus* YM303, a mesophilic strain—to grow and produce cereulide in ice cream mixture and milk under varying temperature conditions. This approach is crucial for understanding the growth behavior and toxin production of different *B. cereus* strains across diverse food matrices.

## 2. Materials and Methods

### 2.1. Strains and Growth Conditions

A total of 57 isolates of *B. cereus* isolated from a Norwegian ice cream production plant were included in this work. Isolates NVH-IS213-231 (*n* = 8), NVH-IS233-241 (*n* = 4), NVH-YM301-330 (*n* = 27), NVH-YM341, NVH-YM345, and NVH-YM357 were isolated from ice cream products, while isolates NVH-YM332-338 (*n* = 3), NVH-YM342, and NVH-YM346-56 (*n* = 9) were isolated from ice cream mass sampled along the production line. NVH-IS232 and NVH-YM359 were isolated from cream used in the production of ice cream. Data of the isolates are listed in Appendix A. Presumptive *B. cereus* was isolated on Mannitol Egg Yolk Polymyxin Agar (MYP) (Oxoid, Basingstoke, UK) according to NMKL (Nordic Committee on Food Analysis) method No. 67 [24]. Affiliation with the *B. cereus* group was confirmed by MALDI-TOF VITEK^®^ MS (bioMérieux, Marcy-l’Étoile, France). Psychrotrophic and mesophilic growth characteristics were assessed on blood agar plates incubated for seven or two days at 7 °C or 43 °C, respectively. Overnight cultures were cultivated in Lysogeny broth (LB) at 30 °C with agitation at 180 rpm. As no psychrotrophic emetic *B. cereus* spp. were identified among the isolates in the present study, we examined the growth characteristics of this group of strains in milk and ice cream mixture using *B. weihenstephanensis* strain BtB2-4, isolated from forest soil in Belgium [19]. NC7401, a well-studied emetic strain from Japan [25], and MC67, a cereulide-producing psychrotrophic strain isolated from sandy loam soil in Denmark [20], were included in the Mashtree analysis.

### 2.2. Genome Sequencing and Phylogenetic Analysis

DNA was extracted according to the method described by Pospiech and Neumann [26]. DNA was quantified using Qubit 3.0 Fluorometer (Invitrogen, Penang, Malaysia), and the DNA integrity was controlled by agarose gel electrophoresis before the DNA was spectrophotometrically controlled for RNA and contaminants using mySPEC (VWR, Wittlich, Germany). The genomic DNA was subjected to library preparation and short-read sequencing using Illumina NovaSeq platform (Illumina, San Diego, CA, USA), followed by raw data filtering consisting of removal of adapters, reads N > 10% (N represents bases that could not be determined), and removal of reads with low-quality (Qscore ≤ 5) base. All steps from DNA quality control in filtering of raw reads were performed by the commercial sequencing provider Novogene Co, UK. Reads were further quality-trimmed using fastp with default settings [27] while genome assembly was carried out by Shovill v1.1.0 (https://github.com/tseemann/shovill, accessed on 15 August 2024). Genome contiguity, completeness, and correctness were assessed using Quast v. 5.0.2 + galaxy3 [28] and BUSCO v. 5.3.2 + galaxy0 (mode genome, gene predictor prodigal, lineage dataset bacillales_odb10) [29]. Species validation was carried out by calculating average nucleotide identity (ANI) against the *B. cereus* reference genome ATCC 14579 (accession number NC_004722.1). The genomes were annotated using the online NCBI Procaryotic Genome Annotation Pipeline (PGAP) [30,31,32], and assemblies are deposited under NCBI BioProject PRJNA1123446.

### 2.3. Typing, Virulome Determination, and Clustering

Mashtree v. 1.1.3 with default settings was used to infer whole genome clustering of the isolates collected in this study [33], including the following reference strains: the cereulide-producing mesophilic strain NC7401 (accession number GCF_000283675.1) from a disease outbreak in Japan [25,34], assembly of *B. weihenstephanensis* strain BtB2-4 (accession number AHDR00000000), and strain MC67 (accession number AHEN01000000). Snippy v4.3.6 was used to identify core SNPs among isolates within a cluster using one random sequence in the cluster as a reference genome [35]. Isolates differing with ≤24 SNPs were considered to belong to the same strain. BTyper3 (https://github.com/lmc297/BTyper3, accessed on 15 August 2024, v3.4.0) was used for taxonomical *panC* classification of *B. cereus* group isolates assigning genomes to a phylogenetic group (Group I-VIII) using an eight-group *panC* group assignment scheme [36,37,38,39]. BTyper3 also performed virulence gene detection including CesNRPS, Nhe, Hbl, and CytK encoding genes [39].

### 2.4. PCR

Presence of the gene encoding cold-shock protein A (*cspA*) was determined by PCR using primers listed in Table 1 [16,40]. PCR reactions were performed using an Eppendorf Mastercycler and DreamTaq DNA polymerase (Thermo Fisher Scientific, Vacaville, CA, USA) according to the manufacturer’s instructions. The following program was used: 2 min denaturation at 95 °C followed by 30 cycles of denaturation at 95 °C for 30 s, annealing at 52 °C for 30 s, and extension at 72 °C for 60 s. A final elongation step of 5 min at 72 °C terminated the program. The PCR products were analyzed on 1% agarose gels using SYBR Safe DNA Gel Stain (Thermo Fischer Scientific).

### 2.5. Growth Curves

Growth curves of *B. cereus* NVH-YM303 and *B. weihenstephanensis* BtB2-4 were established in ultra-pasteurized milk (TINE, Norway) at 8 °C, 15 °C, and 25 °C. Overnight cultures in LB were diluted to 2000 CFU/mL in 10 mL milk in 50 mL plastic tubes (Falcon) and incubated at respective temperatures. After incubation 24 to 312 h, dilutions were plated on LB agar and incubated overnight at 30 °C before counting of colonies.

### 2.6. Cereulide Production in Ice Cream Matrix

Overnight cultures of *B. cereus* in LB were diluted to approximately 2000 CFU/mL in 10 mL ice cream matrix or milk in 50 mL plastic tubes (Falcon). The ice cream matrix consists of skimmed milk, cream (21%), sugar, whey (milk), glucose syrup, dextrose, skimmed milk powder, aroma, emulsifier (mono- and diglycerides of fatty acids), stabilizer (guar gum powder, cellulose gum, carrageenan), and color (carotenes). All experiments involving milk were performed in ultra-pasteurized (UHT) milk containing 3.5% fat from TINE, Norway. The tubes were incubated without agitation with the lid unlocked at 8 °C, 15 °C, 20 °C, 25 °C, or 30 °C. After incubation, the samples were stored at −18 °C until quantification of cereulide.

### 2.7. Quantification of Cereulide by LC-MS/MS

Liquid Chromatography with tandem mass spectrometry (LC-MS/MS) was used as a sensitive method to quantify cereulide levels in milk and ice cream. Sample preparation and LC-MS/MS were performed according to Rønning and coworkers [41]. Most samples were analyzed using an Agilent 1200 SL LC-system (Agilent Technologies, Waldbronn, Germany) coupled with an Agilent G6460 MS/MS (Agilent Technologies, Santa Clara, CA, USA) according to ISO 18465:2017 [42]; however, some samples were analyzed using an Agilent 1260 LC-system (Agilent Technologies, Waldbronn, Germany) coupled with an Agilent G6465B MS/MS (Agilent Technologies, Waghäusel, Germany). Two different LC-MS/MS machines were used during this study because of change to newer equipment. The new equipment was tested before analysis and performed equally well on quality control samples, both before and during analysis of the samples. For both instruments, the limit of detection (LOD) and the limit of quantification (LOQ) were 0.1 ng/g (LOD) and 0.5 ng/g (LOQ) when expected concentration was below 1000 ng/g, and 1.0 ng/g (LOD) and 5.0 ng/g (LOQ) when the expected concentration was up to 3000 ng/g.

### 2.8. Statistics

All descriptive statistics as well as hypothesis testing were handled in Microsoft Office Excel. *p*-values used to test for statistically significant differences between two measurements were calculated using two-tailed paired Student’s *t*-test in Excel. *p*-values equal to or below 0.05 were considered significant. Bar charts and standard deviation were calculated using Excel. Results from experiments determining emetic toxin in milk and ice cream matrix were given as mean of three independent replicates. Growth curves were calculated using means of two independent replicates.

## 3. Results

### 3.1. Genetic Characterization of B. cereus Ice Cream Isolates

A total number of 57 *B. cereus* isolates were collected from a Norwegian ice cream production plant over a period of 19 months (June 2021 to December 2022). The isolates were obtained not only from samples collected during the production process, but also from various types of ice cream products originating from different production lines. A Mashtree distance-based cluster analysis of the genomes of the 57 isolates is shown in Figure 1. The genomes of three strains from external sources including the mesophilic emetic strains NC7401 [25,34] and two psychrotrophic cereulide-producing strains, BtB2-4 [19] and MC67 [20], were included in the Mashtree analysis to set the phylogeny into context with toxigenic characteristics and preferred temperature range.

The isolates were clustered according to *panC* groups, of which 19, 18, and 20 isolates belonged to *panC* group II, III, and IV, respectively. The *panC* III group contained only mesophilic isolates, while *panC* II and IV group were populated with both mesophilic and psychotropic isolates. Within the *panC* groups, the isolates were distributed in separated clusters of highly similar isolates, of which six clusters contained three or more strains (A–F, Figure 1). Of the 57 isolates, a total of 25 distinct strains were detected. For comparison of sequences within one Mashtree cluster (A–F), genome sequences were subjected to core single-nucleotide polymorphism (SNP) analyses using Snippy (Table 2).

Cluster A comprises ten isolates isolated from five different ice cream products, one from cream and four from the ice cream matrix collected along the production line (Table 2). Isolates belonging to cluster A were collected over a period of 16 months, and six were similar to each other and the central representative of cluster A, differing by only 0–3 SNPs. Isolates NVH-IS213 and NVH-YM351 differed with 20 and 24 SNPs against NVH-YM317, respectively. The isolates belonging to cluster A showed no growth at either 7 °C or 43 °C, although they did grow at 10 °C and 42 °C (8 °C and 9 °C were not tested). Cluster B consists of six isolates collected from two different ice cream products originating from two separate production lines over a period of three months (Table 2, Appendix A). Isolates belonging to cluster B differed with 0–8 SNPs against isolate NVH-YM326, which are set as strain representative for cluster B. The isolates of cluster B are psychrotrophic and grew at 7 °C but not at 43 °C. Cluster C consists of four isolates collected from two different ice cream products and one from a sample taken during processing over a period of four months (Table 2). Isolates belonging to cluster C differed with 0–2 SNPs against isolate NVH-YM302, the strain representative for cluster C. Cluster D comprises five isolates collected from two different ice cream products over a period of four months (Table 2). Isolates belonging to cluster D differed with 0–2 SNPs against isolate NVH-YM233, which are set as strain representative for cluster D. Cluster E comprises three isolates isolated from samples collected from the same production line within five days, while cluster F comprises three isolates isolated from two different products over a period of 27 days (Appendix A).

### 3.2. Toxigenic Potential and Temperature Requirements of Isolates

All isolates were tested for the presence of *cspA,* the gene encoding cold-shock protein A, using PCR, and only 2 out of 57 isolates, NVH-IS217 and NVH-IS218, had *cspA*. These two isolates grew at 7 °C (Figure 1) and clustered together with the emetic psychrotrophic *B. weihenstephanensis* strains BtB2-4 and MC67 (Figure 1), indicating that they can be classified as *B. weihenstephanensis*. The six clonal isolates of cluster B (Figure 1) grew at 7 °C, but these isolates did not have *cspA*. Thirty-six isolates, of which nineteen were genetically distinct, were classified as mesophilic as they were able to grow at 43 °C (Figure 1). Notably, thirteen isolates exhibited no growth at either 43 °C or at 7 °C and these isolates were restricted to *panC* group II (Figure 1).

All isolates carried the non-hemolytic enterotoxin (*nheABC*) operon (Appendix A). In total, 32 out of 57 isolates (56%) were positive for *hblCDA* and 18 out of 57 (32%) were positive for *cytK* (Figure 1). Isolates in cluster A were negative for both *hblCDA* and *cytK* (Figure 1). The mesophilic isolates in cluster D, E, and F carried genes encoding all three enterotoxins. Isolates positive for all three enterotoxin types cluster together as do the isolates lacking *cytK* and *hblCDA*. The two *B. weihenstephanensis* strains NVH-IS217 and NVH-IS218 carried *nhe* and *hbl* genes, but not *cytK* (Figure 1).

The emetic isolate NVH-YM303 did not grow on blood agar plates at 7 °C but grew at 43 °C and was therefore considered mesophilic. The isolate is closely related to the mesophilic emetic reference strain *B. cereus* NC7401 [34], and it was the only isolate from the ice cream plant carrying the *ces* operon. Isolate NVH-YM221, which is closely related to NVH-YM303 and NC7401 (Figure 1), did not carry the *ces* operon. The isolates NVH-IS217 and NVH-IS218 are closely related to the cereulide-producing psychrotrophic *B. weihenstephanensis* strains BtB2-4 and MC67. However, in contrast to BtB2-4 and MC67, these two isolates did not carry the *ces* operon (Figure 1).

The cereulide biosynthesis gene cluster (*ces*) of NVH-YM303 was found on contig 2 of 291 kb (submitted NCBI; SAMN41811450). The nucleotide sequence of 27 kB covering the *ces* operon (*cesH*–*cesD*) was 100% identical with a coverage of 100% to the *ces* operon of pNCcld (DDBJ/EMBL/GenBank accession number AP007210) of NC7401 and pCER270 of AH187 (also known as *B. cereus* F4810/72) [34,43].

### 3.3. Growth in Milk

Growth curves of *B. cereus* NVH-YM303 and *B. weihenstephanensis* BtB2-4 were established in UHT milk at 8 °C, 15 °C, and 25 °C (Figure 2).

The growth was comparable between the two strains at 15 °C and 25 °C, but we observed no growth for NVH-YM303 at 8 °C. In contrast, BtB2-4 showed an increase in CFU/mL from 10^3^ to 10^8^ within a time period spanning from 72 to 312 h at a temperature of 8 °C.

### 3.4. Cereulide Production in Milk and Ice Cream Mixture

NVH-YM303 was the only *ces*-positive *B. cereus* strain detected among the 25 distinct isolates characterized in this study (Figure 1). To assess the food safety risk associated with the growth of *B. cereus* spp., in dairy products we examined its production of cereulide in UHT milk and ice cream mixture at 8 °C, 15 °C, 25 °C, and 30 °C (Figure 3A,C). Moreover, for comparison we examined the cereulide-producing capability of the psychrotrophic *B. weihenstephanensis* BtB2-4 at 8 °C, 15 °C, and 25 °C (Figure 3B,D).

We did not detect any cereulide production in milk by NVH-YM303 at 8 °C, even though viable cells were present in the milk for up to 312 h post-inoculation. In contrast, when *B. weihenstephanensis* BtB2-4 were inoculated in milk, it produced on average 910 ± 567 ng/mL between 168 and 240 h at 8 °C. At 15 °C, the levels of cereulide produced in milk were similar between NVH-YM303 and BtB2-4 (608 ± 221 ng/mL and 500 ± 11 ng/mL, respectively) after 168 h of incubation (Figure 3A,B). Notably, both strains had produced considerable amounts of cereulide within 24 h post-inoculation at 25 °C (NVH-YM303 266 ± 44 ng/mL and BtB2-4 151 ± 66 ng/mL).

When grown in ice cream mixture at 25 °C, NVH-YM303 showed the highest concentrations of cereulide measured in this study, reaching concentrations of 1052 ± 14 ng/mL after 48 h and above 1700 ng/mL after 168 h (Figure 3C). At 25 °C, NVH-YM303 produced three times more cereulide in the ice cream mixture compared to what it had produced in milk within a 48 h period (Figure 3A,C). In contrast, at 15 °C NVH-YM303 produced more cereulide in milk than in the ice cream mixture after a period of 168 h (608± 221 ng/mL and 233 ± 10 ng/mL, respectively). Moreover, NVH-YM303 produced a higher amount of cereulide in ice cream mixture after 24 h at 30 °C compared to ice cream mixture at 25 °C (386 ± 34 ng/mL and 175 ± 55 ng/mL, respectively, *p* < 0.01).

NVH-YM303 produced less cereulide in ice cream mixture at 15 °C compared to BtB2-4, but more at 25 °C after 168 h (1724 ± 187 ng/mL vs. 343 ± 251 ng/mL, respectively, *p* < 0.05) (Figure 3C,D). In ice cream mixture at 25 °C, NVH-YM303 began production of cereulide earlier compared to BtB2-4. After 24 h in ice cream mixture at 25 °C, NVH-YM303 generated 175 ± 55 ng/mL of cereulide but BtB2-4 only produced 26 ± 25 ng/mL under the same conditions (*p* < 0.05). BtB2-4 produced cereulide at 8 °C, with the cereulide concentration after 312 h being higher in milk than in ice cream mixture (1787 ± 923 ng/mL and 89 ± 113 ng/mL, respectively) (Figure 3B,D). Since NVH-YM303 did not grow or produce cereulide in milk at 8 °C, cereulide production of this isolate was not tested in the ice cream mixture at this temperature. The production of cereulide within 24 h can serve as an indicator of the strains’ ability to rapidly produce a toxin which is highly relevant for the ice cream industry. In the present study, the highest amounts of cereulide produced within 24 h were detected when NVH-YM303 had grown in milk at 25 °C and in ice cream at 30 °C (266 ± 44 ng/mL and 386 ± 34 ng/mL, respectively) (Figure 3A,C).

## 4. Discussion

We characterized the growth range, enterotoxigenic potential, and genomic relationship of 57 *B. cereus* isolates collected from various points along the production line of a Norwegian ice cream producer over a period of 19 months. Many of the isolates were genomically similar, and as a result, the 57 isolates were distributed among 25 distinct *B. cereus* spp. strains. The isolates which were grouped in one of the six clusters (A–F) (*n* = 31) showed limited intracluster genomic diversity (0–24 SNPs), suggesting that these isolates within a cluster share a recent common ancestral strain [44]. The average number of core SNPs observed between 30 *panC* group III isolates from a foodborne outbreak in 2016 and publicly available group III genomes were more than 300 SNPs [44]. The small number of SNPs observed within clusters in this study strongly suggests that the same strains were isolated at different time points and from various sampling points along the production line, and this indicates that these *B. cereus* strains persist in the production plant over time. Notably, one isolate belonging to cluster A (NVH-YM359) was isolated from cream used in the production process as long as 15 months after the same strain was isolated from an ice cream product. This finding suggests that this strain may be introduced before the cream was brought into the production plant or that the strain may persist in the cream tank.

One isolate (4% of the 25 distinct strains), NVH-YM303, carried the *ces* genes and produced considerable amounts of cereulide. Previous studies indicate a prevalence of 0–4% cereulide-positive strains isolated from vegetables and berries [45,46,47], and a German study from 2008 found that 4.7% of *B. cereus* isolates from ice cream carried the *cesA* gene [48]. The distribution of *B. cereus* enterotoxin genes typically ranges from 85% to 100% for Nhe, from 40% to 70% for Hemolysin BL, and approximately from 40% to 70% for CytK [49]. In this study, all of the 25 distict strains were positive for *nheABC*, 10 were positive for *cytK*, while 13 were positive for two or more of the *hbl* genes which are in line with earlier findings [48].

According to the distance-based cluster analysis, the cereulide-producing strain NVH-YM303 is closely related to the emetic *B. cereus* NC7401, belonging to *panC* group III [36,39]. NVH-YM303 and NC7401 are genetically distant from the two cereulide-producing psychrotrophic *B. weihenstephanensis* strains BtB2-4 and MC67, which belong to group IV [36,39]. Earlier, all emetic *B. cereus* strains were considered closely related and were assigned to a single evolutionary lineage [50]. However, recent studies indicate that cereulide synthetase encoding genes may be acquired or lost in processes that may occur across lineages [51]. This is supported by the results from the current work showing that strain NVH-YM221, which is closely related to the *ces*-positive strains NVH-YM303 and NC7401, did not have *cesA.* Similarly, NVH-IS217 and NVH-IS218, which are closely related to BtB2-4 and MC67, did not carry the *ces* genes. The strong genetic similarity between these strains and the absence of *ces* genes in two of them supports the hypothesis of horizontal gene transfer of the plasmid-borne *ces* genes.

Food matrix is a crucial parameter influencing the activity of the *ces* promoter, and moreover, there is a strong correlation between promoter activity and the amount of cereulide produced [25,52]. Production of cereulide has shown to be higher in farinaceous foods compared to in more proteinaceous food matrices [25,52]. Ice cream mixture is more farinaceous than milk, and we did indeed observe the higher production of cereulide by NVH-YM303 in ice cream compared to in milk kept at 25 °C (1724 ± 187 ng/mL at 168 h and 359 ± 65 ng/mL at 168 h, respectively, *p* < 0.01). However, this dependency on matrix composition seems to be a strain-dependent trait as strain BtB2-4 had a more consistent production of cereulide regardless of food matrix.

Temperature is another important factor that influences the production of cereulide by *B. cereus* spp. [53]. Certain psychrotropic emetic strains can produce cereulide at low temperatures [19,20,22]. Meanwhile, cereulide production by mesophilic emetic strains is, on the other hand, restricted by their growth threshold of 10–15 °C [54,55]. At 8 °C, NVH-YM303 did not grow nor produce cereulide in milk and its production of cereulide in ice cream was considerably lower at 15 °C than at 25 °C. However, in milk at 15 °C, NVH-YM303 and BtB2-4 produced comparable amounts of cereulide. Growth experiments performed in milk showed that NVH-YM303 and BtB2-4 exhibited a slower growth rate at 15 °C than at 25 °C, and that both strains had similar growth curves under these conditions (Figure 2). It is suggested that the optimum temperature for cereulide production lies in the lower part of the range of optimal growth and that the temperature range for cereulide production is more restricted than the temperature range for growth [54,55,56]. In this study, we observe that the mesophilic strain NVH-YM303 produces more cereulide after 24 h at 30 °C compared to at 25 °C in ice cream mixture (Figure 3C). Guérin et al. (2017) observed that the psychrotropic *B. weihenstephanensis* strain BtB2-4 produced the highest amount of cereulide at 25 °C on plate count agar but they did not test cereulide production at 30 °C. In contrast with their report, we observed that in ice cream mixture, BtB2-4 produced less cereulide at 25 °C than at 15 °C, once again emphasizing the significant influence of the food matrix on cereulide production. The genetical distance between NVH-YM303 and BtB2-4 belonging to *panC* group III and IV, respectively (Figure 1), may contribute to differences in the regulation of cereulide production between these two strains. Little is known about the link between genetic distance and the production of cereulide. However, a recent study found that the *ces* genes are highly conserved and that differences in the production of cereulide could not be explained by sequence variations in the *ces* genes or in the regulatory genes *abrB*, *spo0A*, *codY*, and *pagRBc* [57].

The dose of cereulide required to induce emetic symptoms in humans is not precisely defined. However, an analysis of the levels of emetic toxin in foods implicated in episodes of *B. cereus* food poisoning indicate that levels between 0.01 and 1.28 µg/g food are sufficient for the onset of symptoms [25]. According to these observations, the amount of cereulide produced by NVH-YM303 in milk and ice cream mixture (266 ± 44 ng/mL and 175 ± 55 ng/mL, respectively) after incubation at 25 °C for 24 h and the amount of cereulide produced by BtB2-4 in milk after incubation at 25 °C for 24 h (151 ± 66 ng/mL) lie within the range of causing disease.

*B. weihenstephanensis* growing at 4–7 °C but not at 43 °C is also characterized by the presence of the psychrotrophic signature gene *cspA* [15,16]. Two of the isolates from this study, NVH-IS217 and NVH-IS218, grew at 7 °C, carried *cspA*, and were closely related to the two *B. weihenstephaninsis* strains BtB2-4 and MC67. In contrast, six isolates grouped into cluster B, also able to grow at 7 °C, did not contain *cspA*. The presence of *cspA*-negative psychrotrophic *B. cereus* group strains has been described previously and these strains are assigned to *panC* group II, while psychrotrophic strains carrying *cspA* are specific to group VI [36,58]. The *panC* grouping of psychrotrophic strains we see in this study is in line with previous studies [36]. However, ten isolates belonging to cluster A and three additional isolates were neither able to grow at 7 °C nor at 43 °C. These strains were also assigned to *panC* group II. To our knowledge, little is known about *B. cereus* cold adaptation, and studies attempting to identify the genomic mechanism behind psychrophilic growth are therefore of great interest.

We observed very low levels of cereulide (<1.0 ng/mL) in both milk and ice cream after incubation at 24 h at 15 °C following inoculation with either NVH-YM303 or BtB2-4. These findings indicate that maintaining the temperature at or below 15 °C could ensure the lowest possible production of cereulide. However, additional research using different strains that produce cereulide is necessary before any definitive conclusions can be made. Further research is also needed to delve deeper into the complex interplay between various food matrices and *B. cereus* spp. strains, with distinct temperature preferences. This deeper understanding will facilitate the refinement of practices aimed at mitigating bacterial contamination in food, thereby not only elevating food safety standards but also enhancing the microbial quality of food products.

## 5. Conclusions

Our characterization of 57 *B. cereus* spp. isolates from a Norwegian ice cream production facility has provided significant insights into their genomic relationships, growth ranges, and enterotoxigenic potential. The genomic analysis revealed that many isolates share a recent common ancestor, suggesting repeated contamination events from persistent sources.

Of particular concern is the identification of NVH-YM303, a mesophilic isolate carrying a complete cereulide synthetase operon, which poses potential risks associated with toxin production and food poisoning. Our experiments demonstrated that NVH-YM303 produces elevated levels of cereulide in ice cream compared to in milk, with temperature-dependent variations further complicating risk assessments.

## Figures and Tables

**Figure 1 foods-13-03029-f001:**
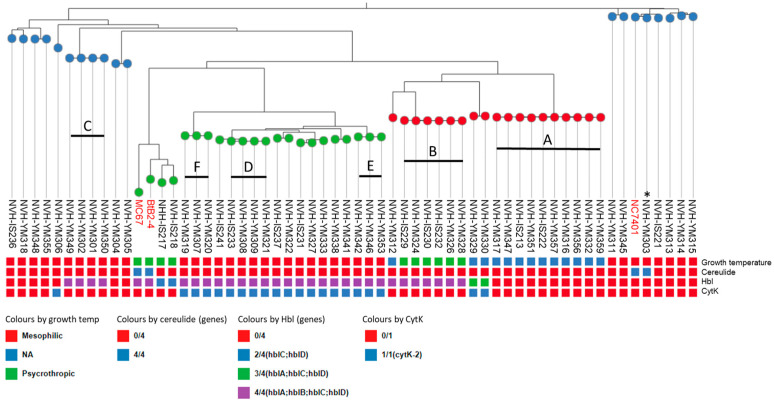
Distance-based Mashtree cluster analysis based on the genomes of 57 *Bacillus cereus*/*B. weihenstephanensis* isolates from a Norwegian ice cream production plant. Nodes are colored according to *panC* groups [38]. Squares are colored according to growth temperature characteristics, presence of CesNRPS genes (cereulide), and presence of the enterotoxin genes *hblCDAB* and *cytK* (see legends). The black lines indicate clusters (≥3 isolates) of highly similar isolates (A–F). Reference strains, one mesophilic emetic *B. cereus strain* (NC7401) and two psychrotrophic *B. weihenstephanensis* strains (BtB2-4 and MC67), are included in the three (in red). Asterisks (*) indicate the cereulide-producing *B. cereus* strain (NVH-YM303) isolated from ice cream. Not applicable (NA) is used for strains that do not grow at either 7 °C or 43 °C. An interactive version of the figure is available at https://microreact.org/project/eRgJS3G1gnrujCzrc5nhoD-ice-cream-bacillus, accessed on 14 June 2024.

**Figure 2 foods-13-03029-f002:**
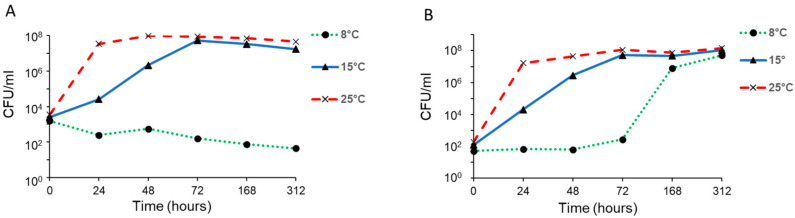
Growth in milk by *B. cereus* NVH-YM303 (**A**) and *B. weihenstephanensis* BtB2-4 (**B**) at 8 °C, 15 °C, and 25 °C for 312 h. The given values are mean of two independent replicates.

**Figure 3 foods-13-03029-f003:**
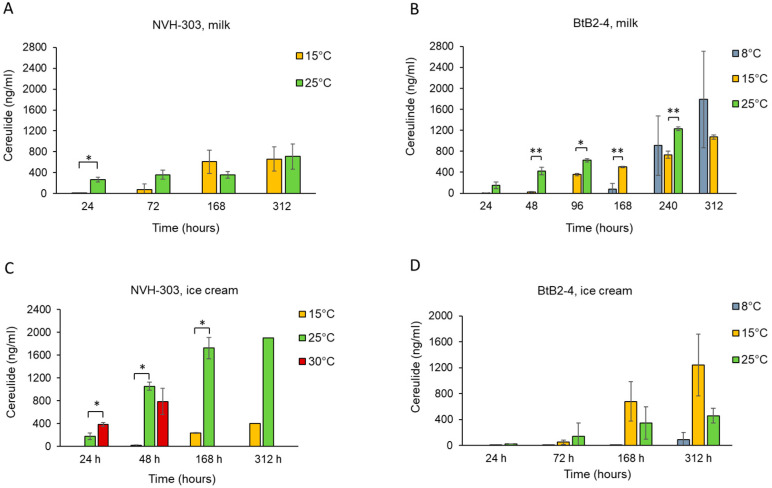
Cereulide production in milk and ice cream by *B. cereus* NVH-YM303 ((**A**) and (**C**), respectively) and by *B. weihenstephanensis* BtB2-4 ((**B**) and (**D**), respectively) at different temperatures. NVH-YM303 did not produce cereulide at 8 °C in milk, and cereulide production by NVH-YM303 in ice cream at 8 °C was not tested. Error bars indicate standard deviations of results obtained from three independent replicates except for two replicates of NVH-YM303 in ice cream at 25 °C/312 h (**C**). Asterisks represent statistical differences from pairwise comparisons determined by two-tailed paired Student’s *t* tests (* *p* ≤ 0.01; ** *p* ≤ 0.05).

**Table 1 foods-13-03029-t001:** Primers used in this study.

Primer	Gene	Sequence	Reference
BcAPF1/BcAPR1	*cspA*	5′-GAGGAAATAATTATGACAGTT-3′/ 5′-CTTYTTGGCCTTCTTCTAA-3′	[16]

**Table 2 foods-13-03029-t002:** Pairwise SNP differences within one Mashtree cluster (A–F) with time of isolation and origin. Each strain was compared to a randomly selected strain in the cluster (reference).

Cluster	Isolate	% Aligned *	SNPs	Date of Isolation	Isolated from
A	NVH-IS213	99.6	20	August 2021	Product a, line 1
	NVH-IS222	98.8	1	June 2021	Product b, line 2
	NVH-YM316	98.8	3	January 2022	Product c, line 2
	NVH-YM332	98.9	2	September 2022	Licm **
	NVH-YM347	99.8	3	September 2022	Licm
	NVH-YM351	99.6	24	September 2022	Licm
	NVH-YM356	99.2	2	September 2022	Freezer
	NVH-YM357	99.1	0	September 2022	Product m
	NVH-YM359	98.9	2	September 2022	Cream
	Reference NVH-YM317 (5,763,304 bp)	100	0	January 2022	Product c, line 2
B	NVH-IS229	99.2	1	October 2021	Product c, line 2
	NVH-IS230	99.9	1	October 2021	Product c, line 2
	NVH-IS232	99.9	8	December 2021	Cream
	NVH-YM324	99.9	0	December 2021	Product c, line 2
	NVH-YM328	99.9	0	December 2021	Product d, line 2
	Reference NVH-YM326 (5,795,337 bp)	100	0	December 2021	Product c, line 2
C	NVH-YM301	100	0	December 2022	Product g, line 1
	NVH-YM349	100	2	September 2022	Pipeline
	NVH-YM350	100	2	September 2022	Licm
	Reference NVH-YM302 (5,324,995 bp)	100	0	December 2022	Product g, line 1
D	NVH-IS241	99.9	11	November 2021	Product f
	NVH-YM308	99.9	0	December 2021	Product c, line 2
	NVH-YM309	99.9	1	March 2022	Product c, line 2
	NVH-YM321	99.9	4	January 2022	Product c, line 2
	Reference NVH-IS233 (5,344,339 bp)	100	0	December 2021	Product d, line 3
E	NVH-YM342	99.7	0	October 2022	Freezer
	NVH-YM353	99.7	1	September 2022	Freezer
	Reference NVH-YM346 (5,691,691 bp)	100	0	September 2022	Freezer
F	NVH-YM307	100	2	December 2021	Product c, line 2
	NVH-YM319	100	1	December 2021	Product h, line 2
	Reference NVH-YM320 (5,706,417 bp)	100	0	January 2022	Product c, line 2

* % aligned to the reference strain within each cluster; ** Licm—leftover ice cream mix.

## Data Availability

The original contributions presented in this study are included in the article/Appendix A; further inquiries can be directed to the corresponding author.

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
