# Peer review of "Genetic Profile and Toxigenic Potential of Bacillus cereus Isolates from a Norwegian Ice Cream Production Plant"

_foods, 2024, doi:10.3390/foods13193029_

Round 1

Reviewer 1 Report

Comments and Suggestions for Authors

The contamination of Bacillus cereus in ice cream factory was studied in this paper, the genetic profile and toxigenic Potential was also discussed. The whole work is interesting and meaningful. There are some questions as follows:

1, A figure of bacterial isolation and culture can be added;

2, Why two different LC-MS/MS equipment were adopted?

3, All the strains were conducted the whole genome sequencing, was the sequence of cspA gene analyzed only through the sequence alignment and mapping? Is the PCR amplification been applied on all the 57 strains?

4, What is the purpose of using LS-MS/MS, not clearly explained, only cited a reference. The same is true elsewhere, the suggestion is that the purpose of each step is clearly described

5, All data was handled in Microsoft Office Excel, is Microsoft Office Excel a professional data processing software?

Comments on the Quality of English Language

English language quality is good

Author Response

Comment 1: A figure of bacterial isolation and culture can be added;

Response 1: A graphical abstract has been added to the revised manuscript (line 24).

Comment 2: Why two different LC-MS/MS equipment were adopted?

Response 2: The equipment that was used in the first analysis was exchanged for newer equipment. The following sentences are added in line 191-193 in the revised manuscript to clarify:

Two different LC-MS/MS machines were used during the study because of change to newer equipment. The new equipment was tested before analysis, and performed equally well on Quality Control Samples, both before and during analysis of the samples.

 Comment 3: All the strains were conducted the whole genome sequencing, was the sequence of cspA gene analyzed only through the sequence alignment and mapping? Is the PCR amplification been applied on all the 57 strains?

Response 3: The whole genome sequences were not analyzed for cspA. However, all 57 strains were analysed for the presence of cspA using PCR.

Comment 4: What is the purpose of using LS-MS/MS, not clearly explained, only cited a reference. The same is true elsewhere, the suggestion is that the purpose of each step is clearly described.

Response 4: To explain better, the following sentence is inserted in line 183-184 in the revised manuscript:

Liquid Chromatography with tandem mass spectrometry (LC-MS/MS) was used as a sensitive method to quantify cereulide levels in milk and ice cream.

Comment 5: All data was handled in Microsoft Office Excel, is Microsoft Office Excel a professional data processing software?

Response 5: The sentence “All data was handled in Microsoft Office Excel” is changed to (line 198 in the revised manuscript):

All descriptive statistics as well as hypothesis testing were handled in Microsoft Office Excel.

Yes, Microsoft Excel is considered a professional data processing software, widely used in biological sciences.

Reviewer 2 Report

Comments and Suggestions for Authors

The study of Toril et al. investigated the genetic identity and growth characteristics of 57 B. cereus isolates collected from a Norwegian ice cream production plant. Besides, they tested the growth and cereulide production of B. cereus strain YM303 in ice cream mixture and milk under various temperature conditions. Overall, the study was interesting and written in good English. The report could benefit readers in the field of food safety. Some minor revisions should be made. 

1、P96-97, Affiliation to the B. cereus group were confirmed by MALDI-TOF-MS, why 16S rDNA sequencing was not conducted. 

2、In figure 2, why there were not data  to show the  CFU number in each time point. 

3、Asterisks to show statistical difference in Figure 3B should be modified properly. 

4、P406-407,Much more description shoud be added how genetical distance contributes to differences in regulation of cereulide production. 

Author Response

Comment 1、P96-97, Affiliation to the B. cereus group were confirmed by MALDI-TOF-MS, why 16S rDNA sequencing was not conducted. 

Response 1: MALDI-TOF was used initially in the study. The taxonomy of the Bacillus genus is complicated at best and 16S rDNA sequence is not accurate enough for this genus (it cannot separate species in the B. cereus group). Therefore two other approaches were used discriminate between isolates: “Species validation was done on by calculating average nucleotide identity (ANI) against the B. cereus reference genome ATCC 14579 (Accession number NC_004722.1)” and typing using BTyper3 by Carrol and colleagues: typing the strains with BTyper3 and assigning strains to different panC groups: “BTyper3 (https://github.com/lmc297/BTyper3) was used for taxonomically panC classification of B. cereus group isolates assigning genomes to a phylogenetic group (Group I-VIII) using an eight-group panC group assignment scheme [36-38]”. This is all described in the manuscript.

Comment 2、In figure 2, why there were not data to show the CFU number in each time point. 

Response 2: Figure 2 has been revised according to reviewer 2 (line 296).

Comment 3、Asterisks to show statistical difference in Figure 3B should be modified properly. 

Response 3: Sorry for the mistake. This has been corrected in the revised manuscript (line 314).

Comment 4、P406-407,Much more description should be added how genetical distance contributes to differences in regulation of cereulide production. 

Response 4: To point out what is known about relationship between genetic distance and cereulide production the following sentences are added to the revised manuscript (line 426-430):

Little is known about the link between genetic distance and production of cereulide. However, a recent study found that the ces genes are highly conserved and that differences in production of cereulide can not be explained by sequence variations in the ces genes or in the regulatory genes abrB, spo0A, codY and pagRBc (56).

Reference 56 has been added to the reference list.

  1. Frentzel, H.; Kraemer, M.; Kelner-Burgos, Y., Uelze L, Bodi, D. Cereulide production capacities and genetic properties of 31 emetic Bacillus cereus group strains. Int. J. Food Microbiol. 2024. 417:110694. doi: 10.1016/j.ijfoodmicro.2024.110694.

Reviewer 3 Report

Comments and Suggestions for Authors

The paper is well written and the research conducted is of great scientific significance in the field of food safety.

Line 37: Normally, the first letter of each amino acid is capitalized.

Line 60: It is preferable to use the term ‘microbiota’ instead of ‘flora’.

Line 72-86: Instead of writing here what they have done and the main result, it would be preferable to write the objectives of the work.

2.1. Strains. This is the place to indicate that the isolates came from a Norwegian ice cream producer, as well as the years of sampling should be indicated (from lines 184-185). All three strains from external sources should be mentioned, not only the strain B. weihenstephanensis strain BtB2-4.

Lines 89-93: Perhaps the total number of strains from each source should be indicated.

Line 95: Please indicate the name of the selective agar for the isolation of presumptive Bacillus cereus.

Author Response

Comment 1: Line 37: Normally, the first letter of each amino acid is capitalized.

Response 1: We are sorry for the mistake. The letters have been capitalized in the revised manuscript (line 38).

Comment 2: Line 60: It is preferable to use the term ‘microbiota’ instead of ‘flora’.

Response 2: Flora is changed to microbiota in the revised manuscript (line 62).

Comment 3: Line 72-86: Instead of writing here what they have done and the main result, it would be preferable to write the objectives of the work.

Reponse 3: We have deleted sentence 72-86 from the original manuscript and instead inserted the following paragraph to clarify the objectives of the study (line 75-85 in the revised manuscript):

The objective of this study was to assess the food safety implications of B. cereus spp. growth in ice cream and dairy products. To achieve this, we collected 57 B. cereus isolates from the processing environment and products of a Norwegian ice cream producer. Through genome sequence analysis, we aimed to evaluate the virulence potential of these isolates and identify any persistent strains within the production facility. Additionally, we explored the temperature preferences of these isolates to determine optimal growth conditions. Furthermore, we tested the ability of two specific B. cereus spp. strains—B. weihenstephanensis BtB2-4, a psychrotrophic strain, and B. cereus YM303, a mesophilic strain—to grow and produce cereulide in ice cream mixture and milk under varying temperature conditions. This approach is crucial for understanding the growth behavior and toxin production of different B. cereus strains across diverse food matrices.

Comment 4: 2.1. Strains. This is the place to indicate that the isolates came from a Norwegian ice cream producer, as well as the years of sampling should be indicated (from lines 184-185). All three strains from external sources should be mentioned, not only the strain B. weihenstephanensis strain BtB2-4.

Response 4: We are sorry for the mistake and have included the following sentences and corrected the number of the reference according to this:

A total of 57 isolates of B. cereus isolated from a Norwegian ice cream production plant were included in this work. (line 104-105 in the revised manuscript)

NC7401, a well-studied emetic strain from Japan (25), and MC67, and cereulide producing psy-chrotrophic strain isolated from sandy loam soil in Denmark (20), were included in the Mashtree analysis.  (line 120-122 in the revised manuscript).

Comment 5: Lines 89-93: Perhaps the total number of strains from each source should be indicated.

Response 5: We have now included the numbers from each source in the manuscript. The sentence now reads: (line 105-108 in the revised manuscript)

Isolates NVH-IS213-231 (n=8), NVH-IS233-241 (n=4), NVH-YM301-330 (n=27), NVH-YM341, NVH-YM345 and NVH-YM357 were isolated from ice cream products, while isolates NVH-YM332-338 (n=3), NVH-YM342 and NVH-YM346-56 (n=9) were isolated from ice cream mass sampled along the production line.

Comment 6: Line 95: Please indicate the name of the selective agar for the isolation of presumptive Bacillus cereus.

Response 6: Mannitol Egg Yolk Polymyxin Agar (MYP) agar (Oxoid) was used for isolation of B. cereus and it has been included in the revised manuscript. (line 111)